# Flexible Film Bulk Acoustic Wave Filter Based on Poly(vinylidene fluoride-trifluorethylene)

**DOI:** 10.3390/polym16010150

**Published:** 2024-01-03

**Authors:** Xiangyu He, Jiaqi Lu, Feng Gao, Shurong Dong, Juan Li, Hao Jin, Jikui Luo

**Affiliations:** 1College of Information Science & Electronic Engineering, Zhejiang University, Hangzhou 310027, China; 22160571@zju.edu.cn (X.H.); jiaqilu@zju.edu.cn (J.L.); hjin@zju.edu.cn (H.J.); jackluo@zju.edu.cn (J.L.); 2ZJU-Hangzhou Global Scientific and Technological Innovation Center, Zhejiang University, Hangzhou 310027, China; 3College of Materials Science and Engineering, Zhejiang University of Technology, Hangzhou 310014, China; juanli@zjut.edu.cn

**Keywords:** P(VDF-TrFE), film bulk acoustic resonators, flexible electronics, RF filters

## Abstract

Poly(vinylidene fluoride-trifluorethylene) (P(VDF-TrFE)) has promising potential applications in radio-frequency filters due to their excellent piezoelectric properties, flexibility, and stability. In this paper, a flexible film bulk acoustic wave filter is investigated based on P(VDF-TrFE) as piezoelectric film. A new method based on three-step annealing is developed to efficiently remove the porosity inside the P(VDF-TrFE) films so as to improve its properties. The obtained film achieved high β-phase content beyond 80% and a high piezoelectric coefficient of 27.75 pm/V. Based on the low porosity β-phase films, a flexible wide-band RF filter is designed, which consists of a bulk acoustic wave resonator and lumped inductor-capacitor elements as a hybrid configuration. The resonator sets the filter’s center frequency, while the lumped LC-based matching network extends the bandwidth and enhances out-of-band rejection. The testing results of the proposed wide-band filter show its good performance, with 12.5% fractional bandwidth and an insertion loss of 3.1 dB. To verify the possibility of folding and stacking the flexible bulk acoustic wave devices for high-density multi-filter integration in MIMO communication, bending tests of the filter are also conducted with the bending strain range up to 5500 με. The testing results show no noticeable performance degradation after four bending cycles. This work demonstrates the potential of β-phase P(VDF-TrFE) bulk acoustic wave filters to expand the scope of future flexible radio-frequency filter applications.

## 1. Introduction

Film bulk acoustic wave (BAW) filter is one of the core components for radio frequency (RF) front-end systems such as 5G, and WIFI-6E [1,2,3] due to their ultra-compact size, low insertion loss (IL), and high power-handling capability [4,5]. Despite the widespread application of BAW filters, two hurdles are yet to be overcome. One is the implementation of BAW filters with ultra-wide bandwidth. The n79 band in 5G applications needs a 150–200 MHz bandwidth, while new protocols such as WIFI 6E even require a bandwidth of 2 GHz centered at 6 GHz. It is very difficult to achieve such wide bandwidth using conventional BAW filters [6,7]. Increasing the substrate’s effective electro-mechanical coupling coefficient (*k_t_*^2^) is an effective method to boost the filter bandwidth [8]. Most BAW filters use aluminum nitride (AlN) Scandium (Sc) doped AlN and LiNbO_3_ as the piezoelectric films, but they have a *k_t_*^2^ of 6%, 9%, and 11%, respectively [9]. Thus, they are not good enough for fabricating BAW filters with sufficient bandwidth. Poly(vinylidene fluoride) (PVDF) is the material with the highest *k_t_*^2^, in the range between 14 and 25% [10]. Therefore, it could offer potential applications in future broadband filters. Another hurdle is the integration density of filters. With the emergence of technologies such as multi-channel communication and MIMO, more and more filters are required in the RF front-end modules. For example, modern 5G mobile phones need 60 to 70 BAW filters, which occupy a large area. To reduce the filter size, flexible BAW filters that can be folded and stacked for vertical integration are extremely desirable, but yet to be developed. Conventional BAW filters based on inorganic piezoelectric materials cannot meet such a requirement because of the rigid nature of the substrates. In contrast, piezoelectric polymers such as PVDF and the co-polymer P(VDF-TrFE) are good candidates for such applications.

PVDF is a high-performance piezoelectric polymer known for its remarkable characteristics, including flexibility, good piezoelectric properties, and excellent stability. Compared to PVDF, P(VDF-TrFE) offers higher *k_t_*^2^ [11,12]. The introduction of TrFE increases the unit cell size and the ferroelectric phase’s inter-planar distance of PVDF. It also improves the crystallinity of the material. Both of them enhance the *k_t_*^2^ value of the P(VDF-TrFE) films. In recent years, P(VDF-TrFE) has been explored for the fabrication of BAW devices. In 2018, Ryosuke et al. fabricated a P(VDF-TrFE)-based BAW transducer with a resonant frequency of 387 MHz [13]. In 2020, Wu et al. developed a flexible BAW sensor with a 100-µm thick P(VDF-TrFE) film [14]. Although P(VDF-TrFE) is an excellent candidate for the fabrication of flexible BAW filters due to its high *k_t_*^2^ and flexibility, such P(VDF-TrFE) BAW filters with high performance are yet to be developed.

To explore the idea of high-density foldable integration of multiple high bandwidth filters, this paper reports on the development of flexible P(VDF-TrFE) BAW RF filters with operating frequencies of 13, 50, and 145 MHz, respectively. We have proposed a process for thin film preparation by introducing a three-step annealing method to achieve a denser and smoother film, which significantly enhanced the piezoelectric performance of the film. The resonators constructed with the P(VDF-TrFE) film achieved a *k_t_*^2^ of up to 17. The filter also showed a high fractional bandwidth (FBW) of 12.5%. The P(VDF-TrFE) BAW filter underwent cyclic bending tests with an applied strain up to 5500 με and exhibited no noticeable performance deterioration. This work demonstrated the potential of using P(VDF-TrFE) BAW devices for flexible and foldable high-density high-bandwidth RF filters.

## 2. P(VDF-TrFE) Film Fabrication

### 2.1. Preparation of P(VDF-TrFE) Thin Films

The synthesis process for preparing the P(VDF-TrFE) thin films is schematically shown in Figure 1. P(VDF-TrFE) powder (Piezotech Arkema Co., Stockholm, Sweden PVDF/TrFE = 70/30 mol %, Mw = 450,000 g/mol) was dissolved in dimethylformamide (DMF) (Shanghai Aladdin Biochemical Technology Co., Ltd., Shanghai, China) in a volume ratio of 1:10 assisted with magnetic stirring at room temperature for 6 h to ensure the complete dissolution of P(VDF-TrFE) within the DMF solvent. Then, the P(VDF-TrFE) solution was spin-coated on a silicon wafer coated with a 200 nm copper (Cu) sacrificial layer which was deposited by sputtering. Subsequently, the wafer was placed in an oven at 60 °C for evaporation of the DMF solvent. Following the solvent evaporation, annealing was conducted to produce β-phase crystallinity of the P(VDF-TrFE) film.

In our previous work, an approach involving two-step annealing and surface etching was employed to effectively eliminate pores within P(VDF-TrFE) films and reduce surface roughness [15]. Since the annealing process has significant influences on P(VDF-TrFE) properties, including the content of β-phase, piezoelectric coefficient, and porosity, a novel three-step annealing process (sample S1) has been developed in this work, achieving good flatness and high density without the need for surface etching and simplifying the fabrication process. It features a first annealing step at 160 °C for 30 min, a second step at 180 °C for 20 min, and a final step at 150 °C for 60 min. The first step was introduced to allow the slow evaporation of the DMF solution on the film surface, preventing rapid evaporation and pore formation at 180 °C. For comparison, the traditional constant-temperature annealing process (sample S2) was also conducted on the control sample at 180 °C for 80 min. The above annealing processes were both carried out in a vacuum environment. After annealing, direct current polarization was applied to the film with an electric field strength of up to 150 MV/m for 1 h at room temperature. Finally, the samples were immersed in a FeCl_3_ solution to dissolve the Cu layer and release the P(VDF-TrFE) thin films.

### 2.2. Characterization of P(VDF-TrFE) Thin Films

Gold nanoparticles were coated using an ion sputtering instrument (KYKY SBC-12, Beijing, China ), and the surface morphology and cross-section were measured using field emission scanning electron microscopy (FE-SEM) (Hitachi SU5000, Marunouchi 1-chome, Chiyoda ku, Tokyo, Japan). Figure 2 shows SEM images of various morphological structures of the P(VDF-TrFE) thin films obtained using different annealing processes. Compared with sample S2 (Figure 2b, conventional annealing), sample S1 (Figure 2a, three-step annealing) exhibits lower porosity and a more compact structure. Sample S2 has many voids due to the rapid evaporation of residual DMF solvent at 180 °C as DMF’s boiling point is 153 °C. Conversely, internal voids and cracks in sample S1 were effectively eliminated [16]. The surface structure of P(VDF-TrFE) films was characterized using a multifunctional X-ray diffractometer (XRD) (Shimadzu LabX XRD-6100, Zuokyo ku, Kyoto, Japan). X-ray diffraction (XRD) results of the P(VDF-TrFE) thin films in Figure 2c also show that S1 has a higher full width at half maximum (FWHM) of the spectrum, obtained using a Nicolet 5700 instrument (Thermo Electron Scientific Instruments, Madison, WI, USA), indicating the higher crystallinity of the sample. Figure 2d presents a comparison of FTIR spectra from the P(VDF-TrFE) and PVDF thin films, which provides insights into the crystalline phase contents of materials. The distinctive absorption band at 844 ^−1^ identifies the electrically active and polar β-phase, while the absorption at 763 ^−1^ represents the non-polar α-phase [17]. The specific content *F(β)* of the β-phase can be calculated using the following formula:(1)$$F\left(\beta \right)=\frac{A_{\beta }}{\left(K_{\alpha }/K_{\beta }\right)A_{\alpha }+A_{\beta }}$$
where *A_α_* and *A_β_* represent the absorption rates of the α phase and the β phase at 763 cm^−1^ and 844 cm^−1^, respectively. *K_α_* and *K_β_* are the absorption coefficients at their respective wavenumbers. The values 6.1 × 10^4^ cm^2^ mol^−1^ and 7.7 × 10^4^ cm^2^ mol^−1^ are for *K_α_* and *K_β_* [18], respectively. Upon calculation, it is observed that the β-phase content in sample S1 is approximately 80%, significantly exceeding the 51% content in sample S2. Consequently, the three-step annealing method proves effective in achieving a higher β phase content in the thin film. This can be attributed to the method’s ability to mitigate the evaporation of DMF solvent from the film surface, thereby reducing the presence of surface voids and subsequently enhancing the film’s crystallinity.

The piezoelectric performance was evaluated using a piezoresponse force microscope (PFM, Bruker ICON, Saarbrucken, Germany). The Piezoresponse Force Microscope (PFM) is designed based on the inverse piezoelectric effect, where an alternating current (AC) voltage is applied to the probe, resulting in a periodic deformation of the sample, leading to a cyclic variation in the probe’s height. The P(VDF-TrFE) thin films were polarized before PFM measurement. The electric field applied to polarize the sample must be at least double the coercive field (50 MV/m) of the sample [19]. Due to the risk of air breakdown, the field applied during measurements was significantly lower than the coercive field, making it challenging to attain the d33 hysteresis curve for the extraction of piezoelectric parameters. In experiments, when an electric field greater than 50 MV/m was applied to sample S2, the thin film was prone to breakdown. In contrast, sample S1 was able to be stably polarized under an electric field of up to 250 MV/m. Figure 2e shows the complete valid butterfly curve obtained for sample S1, from which a d33 value of 27.73 pm/V can be obtained. The red and blue lines represent two different scanning directions. As depicted in Figure 2f, the phase contrast in PFM reveals a 180-degree phase difference in both the scanning directions within the piezoelectric loop, indicating the good ferroelectric properties of the P(VDF-TrFE) thin film [20].

## 3. P(VDF-TrFE) BAW RF Filters

### 3.1. Design and Simulation of High k_t_^2^ P(VDF-TrFE) BAW Resonator

Conventional longitudinal mode BAW is made of a piezoelectric layer sandwiched by two metal electrodes. The piezoelectric film converts sinusoidal electrical signals into mechanical vibrations, which form a standing wave in the film with a frequency determined by the film thickness [21]. Finite element analysis (FEA) was utilized to simulate P(VDF-TrFE) BAW. Figure 3a shows a typical longitudinal BAW configuration with a top and a bottom electrode. The thickness of the P(VDF-TrFE) film is 20 μm, while the two aluminum (Al) electrodes are both 0.2 μm in thickness. Since the thickness of the Al electrodes can be neglected as compared to the P(VDF-TrFE) film thickness, the resonant frequency of the device can be estimated by the following formula,
(2)$$f=\frac{v}{2d}$$
(3)$$k_{t}^{2}=\frac{\pi^{2}}{4}\left(\frac{f_{p}-f_{s}}{f_{p}}\right)$$
where *v* represents the longitudinal acoustic velocity of the piezoelectric layer, and *d* represents the thickness of the piezoelectric layer. $f_{s}$, and $f_{p}$ are the resonator’s impedance and serial resonant frequency, respectively. The *k_t_*^2^ is a major performance metric to describe the coupling strength between the electrical and mechanical energy of an acoustic resonator and determines the bandwidth of the resonator-based filter [22]. The simulated typical impedance curve of the BAW with a 20 μm P(VDF-TrFE) film is shown in Figure 3b. The *k_t_^2^* obtained is calculated to be 18%, which shows a large effective electromechanical coupling coefficient for the P(VDF-TrFE) BAW.

The material properties of P(VDF-TrFE) film used for the simulations are shown in Table 1, which describes typical experimental results for piezoelectric, dielectric, and elastic compliance constants. The dielectric loss ($\tan\delta_{\varepsilon }$) of P(VDF-TrFE) film was assumed to be 0.1 [23]. In addition, the driving voltage was all set to 1 V in the FEA modeling. Fixed boundary conditions were applied to the left and right sides of the structure, which is equivalent to the actual device structure with the film edge anchored to its supporting substrate. To include the edge-damping effect and wave adsorption by the supporting substrate, the final 5 μm of the film was set to a perfect matching layer (PML) on the left and right edges of this structure. As mesh sparsity affects the simulation accuracy, mesh optimization by checking the convergence of the results under different mesh densities was also performed to balance the computation time and modeling accuracy.

### 3.2. Fabrication and Testing of High k_t_^2^ P(VDF-TrFE) BAW Resonators

The stability of P(VDF-TrFE) is limited to temperatures below 60 °C, as temperatures exceeding this threshold lead to changes in the film’s domain structure, resulting in a decrease in its piezoelectric performance. Consequently, conventional curing temperatures are unsuitable for P(VDF-TrFE). Moreover, P(VDF-TrFE) lacks chemical resistance to acetone, which restricts the use of a lift-off process. Therefore, we employ a shadow mask for the fabrication of the top and bottom electrodes of FBAR. The utilization of a hard mask fabrication process not only reduces device manufacturing costs but also mitigates potential damage to the film’s properties associated with processes such as photolithography [24].

As depicted in Figure 4a, the fabricated P(VDF-TrFE) BAW resonator comprises a P(VDF-TrFE) piezoelectric film sandwiched by two Al electrodes, which is made by electron beam evaporation. The thicknesses of the two electrodes are both 200 nm. Three different thicknesses of 10, 20, and 50 μm were used for fabricating BAW resonators with different operating frequencies. These thicknesses were chosen because they can achieve good transparency and flexibility while maintaining adequate mechanical strength. The shape of the BAW resonators is an irregular pentagon, which can effectively suppress the generation of lateral spurious modes. Figure 4b shows an image of the fabricated BAW resonator. Figure 4b–d illustrate the impedance spectra of the three BAW resonators with different P(VDF-TrFE) thicknesses.

Resonant frequency, anti-resonant frequency, and coupling coefficient *k_t_*^2^ were extracted from the impedance curves and summarized in Table 2. The resonant frequency of the BAW resonators decreases with an increase in P(VDF-TrFE) thickness. All the devices show a typical resonant frequency (fs) and anti-resonant frequency (fp), as shown in Figure 4b–d, but the device with a 20 μm P(VDF-TrFE) film exhibits the highest *k_t_*^2^ of 17. The peak sharpness of the impedance curve is the indication of the resonance strength of the BAW. Figure 4d reveals that the peak amplitude for the 70 μm P(VDF-TrFE) film BAW is considerably sharper as compared to those of the 10 μm and 20 μm P(VDF-TrFE) film BAWs. This phenomenon is attributed to the presence of more surface air gaps in thinner films during the polarization process, which makes them more susceptible to breakdown under applied electric fields, thus resulting in incomplete polarization. This phenomenon adversely affects the performance of P(VDF-TrFE) resonators due to lower piezoelectric coefficients.

### 3.3. Wide Bandwidth P(VDF-TrFE) RF Filters

The proposed filter configuration is shown in Figure 5a, which includes a P(VDF-TrFE) BAW resonator, a parallel inductor L0, and matching networks on both sides. Figure 5b displays a photo of the fabricated filter. In this filter topology, the BAW resonator and L0 exhibit sharp roll-off characteristics, defining the filter’s center frequency. Figure 6a shows the filtering characteristics of this topology circuit. The matching network consists of a series inductor Ls, a parallel inductor Lp, and a parallel capacitor Cp, resulting in two symmetrical transmission zeros, as illustrated by the blue line in Figure 6b [24]. Essentially, it forms an LC bandpass filter, providing the filter with a wide bandwidth. As depicted in Figure 4b, the assembly process for the filter commences with the design of the flexible PCB, involving dimension planning and surface cleaning. The dimensions of the BAW filter were determined by the fabricated BAW resonator, followed by its integration with the flexible printed circuit board (PCB) containing the matching network. The PCB was fabricated using a flexible substrate with a thickness of 0.11 mm. The dimensions of the manufactured filter measure at 6.3 cm × 2.16 cm, as shown in Figure 5b.

Figure 6b illustrates that the filter is centered around the resonant frequency (fs) of the BAW resonator and exhibits a wide bandwidth with two deep transmission zeros (TZ) symmetrically positioned. The inclusion of a parallel inductor L0 serves to decouple and anti-resonate while generating two symmetrical zeros, as shown by the red line in Figure 6b. The flexibility of the filter topology in achieving high-performance characteristics is demonstrated by adjusting the block components within the matching network. Finally, by combining the two topologies, the proposed filter is realized, which achieved a high bandwidth of 13 MHz and an insertion loss (IL) of 2.87 dB.

### 3.4. Foldability of the BAW filter

The P(VDF-TrFE) BAW resonator’s performance under strain-induced bending was investigated using a series of bending experiments. A mechanical arm (HSV-500, Qingdao TUOKEInstrument Co., Ltd., Qingdao, China) was employed to control the applied strain, while a network analyzer was utilized to measure the corresponding frequency response. Figure 7a illustrates the return loss (S11) spectra of the device under varying levels of strains. As the bending strain increases from zero to 4500 με, the resonance frequency shifts from 50.36 MHz to 49.75 MHz. The BAW resonator exhibits robust performance over a broad strain range, demonstrating the exceptional flexibility of the P(VDF-TrFE) BAW resonators. One potential factor leading to nonlinearity is the alteration of intermolecular spacing and crystal structure of P(VDF-TrFE) polymer under the influence of stress. It has a good crystal structure, which is responsible for its piezoelectric activity and elevated longitudinal sound velocity.

The fabricated BAW filters were subjected to measurements under different strains at room temperature. The outcomes are illustrated in Figure 7b, featuring distinguished filter characteristics, including a low insertion loss (IL) of 3.1 dB, a broad fractional bandwidth (FBW) of 12.5%, and substantial out-of-band attenuation of 15 dB. As shown in Figure 7b, the filter was tested for bending deformation four times, and as the strain gradually increased, the insertion loss of the filter decreased from 3 dB to 6 dB, with a slight fluctuation in the in-band, but the out-of-band rejection is largely unaffected. The BAW filter demonstrates the capability for flexible bending and folding under strains of up to 5500 με. Figure 7c shows the bending and folding of the filter when a strain of about 4000 με was applied to the filter, further illustrating the superb stability of this filter under a variety of strains and showing its potential for application in flexible and foldable filter devices.

## 4. Conclusions

This study encompasses the analysis, design, fabrication, and characterization of flexible P(VDF-TrFE) Film BAW filters. A novel fabrication process for high-quality P(VDF-TrFE) films was developed based on a three-step annealing procedure. The obtained P(VDF-TrFE) film exhibits low porosity and achieves a high β-phase content exceeding 80%, along with a high piezoelectric coefficient of 27.75 pm/V. Flexible BAW devices were manufactured using these low-porosity P(VDF-TrFE) films, resulting in strong resonance with a high K_t_^2^ of 17%, much higher than resonators fabricated with aluminum nitride (AlN), scandium (Sc)-doped AlN and LiNbO_3_.

Furthermore, a filter topology incorporating a P(VDF-TrFE) BAW was proposed and investigated, demonstrating exceptional performance characterized by low insertion loss (IL), broad bandwidth, and high out-of-band suppression. Simulated filter characteristics were experimentally validated. Throughout the entire process, from resonator fabrication to the final filter assembly, the presented flexible Film BAW filter utilizing P(VDF-TrFE) exhibited a substantial fractional bandwidth (FBW) of 12.5%. It underscores the formidable potential for addressing the future demand for ultra-wide band filters. Due to its remarkable flexibility, this device performed effectively within a wide bending strain range of up to 5500 με. The proposed flexible Film BAW filter holds significant potential for applications in foldable integrated flexible acoustic wave devices.

## Figures and Tables

**Figure 1 polymers-16-00150-f001:**
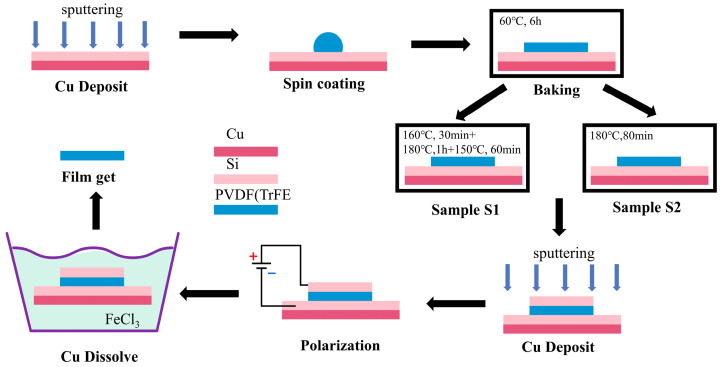
The fabrication process of the P(VDF-TrFE) thin films.

**Figure 2 polymers-16-00150-f002:**
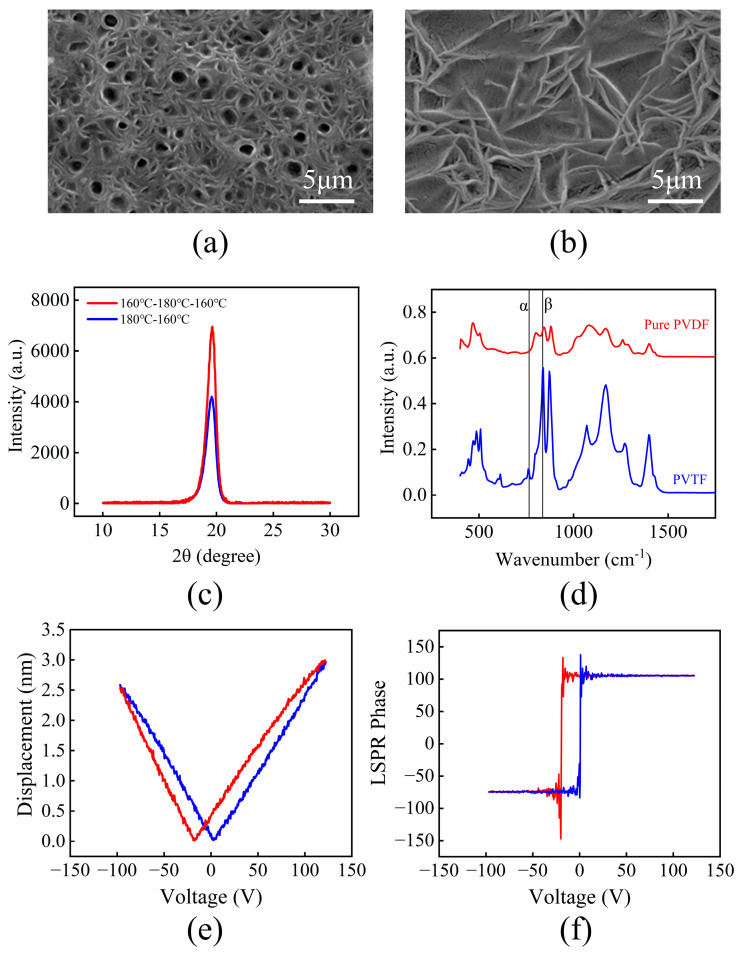
P(VDF-TrFE) Thin Films: (**a**) sample S1 cross-section; (**b**) sample S2 cross-section (**c**) XRD curves of sample S1 and S2; (**d**) FTIR spectra of sample S1 and S2; (**e**) amplitude versus applied voltage of piezoresponse hysteresis loops in P(VDF-TrFE) films; (**f**) phase versus applied voltage.

**Figure 3 polymers-16-00150-f003:**
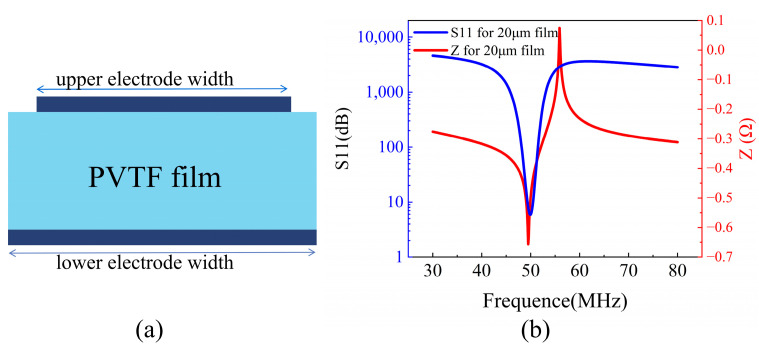
(**a**) Schematic diagram of the structure of the P(VDF-TrFE) BAW Resonators; (**b**) simulated S11 and Z curves of P(VDF-TrFE) BAW with 20 μm P(VDF-TrFE) film.

**Figure 4 polymers-16-00150-f004:**
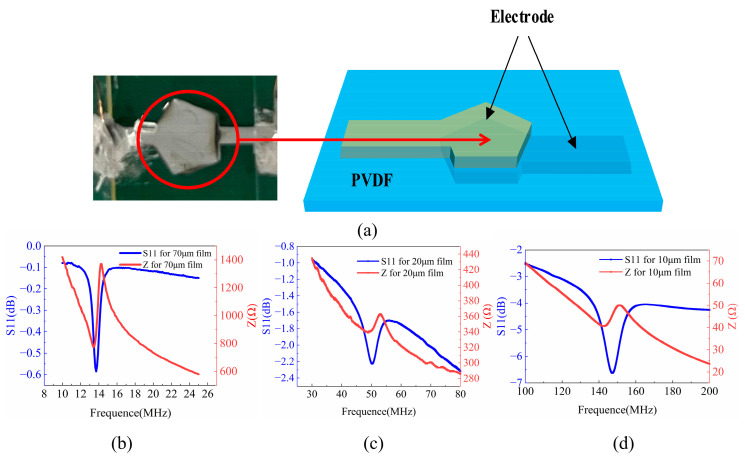
(**a**) A photo and drawing of the fabricated P(VDF-TrFE) BAW; S11 and Z spectra of (**b**) the BAW with a 70 μm P(VDF-TrFE) film; (**c**) with a 20 μm P(VDF-TrFE) film; (**d**) with a 10 μm P(VDF-TrFE) film, respectively.

**Figure 5 polymers-16-00150-f005:**
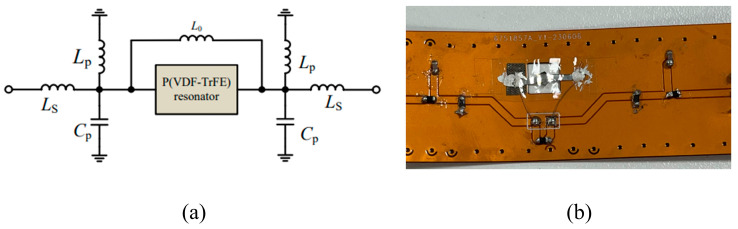
The proposed P(VDF-TrFE) BAW filter. (**a**) Circuit diagram for the proposed BAW filter; (**b**) a photo of the fabricated filter.

**Figure 6 polymers-16-00150-f006:**
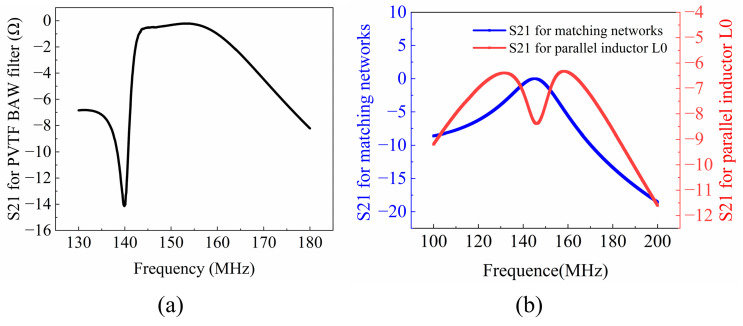
(**a**) The frequency response of fabricated P(VDF-TrFE) BAW filter; (**b**) matching networks and the parallel inductor.

**Figure 7 polymers-16-00150-f007:**
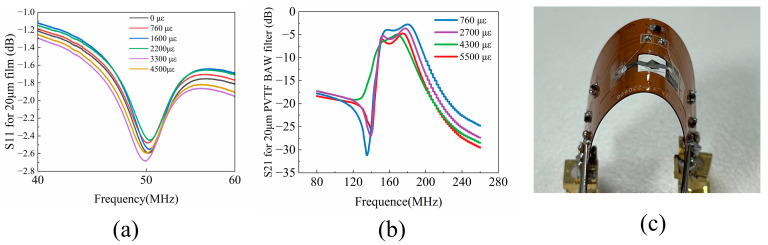
The flexible P(VDF−TrFE) BAW filter (**a**) performance of resonator under different strains; (**b**) performance of filter under different strains; (**c**) bending testing.

**Table 1 polymers-16-00150-t001:** Elastic, piezoelectric, and dielectric matrix of the P(VDF-TrFE).

Elastic compliance [×10^−10^ m^2^/N]	$s_{tj}^{E}=\left[\begin{array}{cc} \begin{array}{ccc} 3.32 & -1.44 & -0.89\\ -1.44 & 3.24 & -0.86\\ -0.89 & -0.86 & 3.00 \end{array} & \begin{array}{ccc} 0 & 0 & 0\\ 0 & 0 & 0\\ 0 & 0 & 0 \end{array}\\ \begin{array}{ccc} 0 & 0 & 0\\ 0 & 0 & 0\\ 0 & 0 & 0 \end{array} & \begin{array}{ccc} 94.0 & 0 & 0\\ 0 & 96.3 & 0\\ 0 & 0 & 14.4 \end{array} \end{array}\right]$
Piezoelectric d constants [pC/N]	$d_{nj}=\left[\begin{array}{ccc} 0 & 0 & 0\\ 0 & 0 & 0\\ 10.7 & 10.1 & -33.5 \end{array}\begin{array}{ccc} 0 & -36.3 & 0\\ -40.6 & 0 & 0\\ 0 & 0 & 0 \end{array}\right]$
Dielectric constant	$\delta_{nm}=\left[\begin{array}{ccc} 7.40 & 0 & 0\\ 0 & 7.95 & 0\\ 0 & 0 & 7.90 \end{array}\right]$

**Table 2 polymers-16-00150-t002:** The performance of resonators with different thicknesses of P(VDF-TrFE) films.

PVTF Thickness /μm	Resonant Frequency (fs)/MHz	Anti-Resonant Frequency (fp)/MHz	Effective Electromechanical Coupling Coefficient
10	143.5	151.1	11.7%
20	49.99	54.19	17%
70	13.38	14.20	13.4%

## Data Availability

The data presented in this study are available upon request from the corresponding author.

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
