# Peer review of "Flexible Film Bulk Acoustic Wave Filter Based on Poly(vinylidene fluoride-trifluorethylene)"

_polymers, 2024, doi:10.3390/polym16010150_

Round 1
Reviewer 1 Report
Comments and Suggestions for Authors
The papers contains a very detailed and complete description of the preparation of interesting products with good performances in sensors application.
It however looks more as an improvement of previous work from the same group of authors rather than a completely new scientific paper offering new inventive concepts. In particular I have the impression , the authors can demonstrate that I am wrong, that the basic concepts and fundamental results of the present paper are similarly reported in the first part of reference (23) of the present ms. The preparative procedure reported in the two papers as Figure 1 is an example, but also other figures and concepts.
Thus I would ask the authors to reconsider these comments and revise the present paper by introducing a clear comparison with reference 23 in order to demonstarte to the reader the real innovation developped.
Comments on the Quality of English LanguageModerate revision looks sufficient
Reviewer 2 Report
Comments and Suggestions for Authors
1-The study has good results and the content of the paper is valuable.
2- Symbol such as PVTF must define for first uses.
3- Introduction need to enlarge by mention more recent achievement in this work.
4- Purity and source of all used materials must mention.
Reviewer 3 Report
Comments and Suggestions for Authors
The manuscript entitled "Flexible Film BAW Filter Based on P(VDF-TrFE)" is a valid research work with appropriate level of novelty and originality. The introduction section clearly presents the state of the art in the specific field of the research which is appropriately substantiated by the relevant references including the recent ones. The presented results are described and treated mostly correctly. The conclusion section adequately summarizes the results. This work can have definite impact on the specific field of the research. Here are the specific comments:
(A) The title should be revised - no abbreviations please.
(B) Authors too broadly using the abbreviations. In some cases they are not clearly described. Please remove ALL the abbreviations from the abstract. Please clearly describe each abbreviation in the body of the manuscript at the place of its first appearance. This is a technical but crucial comment - should be corrected. This will improve the overall clarity of the manuscript.
(C) The source and purity of all used chemicals should be clearly described. All the used devices/techniques (SEM, FTIR, XRD, PFM) should be fully described (model, company, tow, country) as well as the respective samples/conditions used for the measurement. This will contribute to better reproducibility of the results.
(C) The scale bar on figure 2a,b is almost invisible, please improve.
(D) Figure 4 a is too small, please increase the size.
(E) Please slightly extend the conclusion section and explicitly stress the novelty of the obtained results with respect to the state of the art.
This manuscript can be considered for acceptance ONLY after a careful revision.
Round 2
Reviewer 1 Report
Comments and Suggestions for Authors
The authors response is on my viewpoint adequate and justify the proposed new paper for publication. Clearly the innovative aspects described in the letter , and in the revised ms also made clerer, are very specialized and the paper is therfore directed to real specilaists in the field.
In any case I feel confident that it deserves to be accepted for publication.
Reviewer 3 Report
Comments and Suggestions for Authors
Authors considerably improved the manuscript in accordance with reviewer's comments. The manuscript can be considered for acceptance at Polymers as it is.